# LJELSR: A Strengthened Version of JELSR for Feature Selection and Clustering

**DOI:** 10.3390/ijms20040886

**Published:** 2019-02-18

**Authors:** Sha-Sha Wu, Mi-Xiao Hou, Chun-Mei Feng, Jin-Xing Liu

**Affiliations:** 1School of Information Science and Engineering, Qufu Normal University, Rizhao 276826, China; wushashaqf@gmail.com (S.-S.W.); mixiaohousz@gmail.com (M.-X.H.); strawberry.feng0304@gmail.com (C.-M.F.); 2Bio-Computing Research Center, Harbin Institute of Technology, Shenzhen 518055, China

**Keywords:** differentially expressed genes, feature selection, L_1_-norm, sample clustering, sparse constraint

## Abstract

Feature selection and sample clustering play an important role in bioinformatics. Traditional feature selection methods separate sparse regression and embedding learning. Later, to effectively identify the significant features of the genomic data, Joint Embedding Learning and Sparse Regression (JELSR) is proposed. However, since there are many redundancy and noise values in genomic data, the sparseness of this method is far from enough. In this paper, we propose a strengthened version of JELSR by adding the L_1_-norm constraint on the regularization term based on a previous model, and call it LJELSR, to further improve the sparseness of the method. Then, we provide a new iterative algorithm to obtain the convergence solution. The experimental results show that our method achieves a state-of-the-art level both in identifying differentially expressed genes and sample clustering on different genomic data compared to previous methods. Additionally, the selected differentially expressed genes may be of great value in medical research.

## 1. Introduction

With the emergence of deep sequencing technologies, considerable genomic data have become available. Since genomic data are usually high-dimension small-sample data, that is, the dimension of the gene is large, the dimension of the sample is small, and it is easy to cause interference when performing feature selection and difficult to understand the sample directly [1]. Additionally, a large number of superfluous and extraneous genes are possessed in these genomic data, which severely interfere with the biological processes. As the case stands, only a small minority of genes with biological sense contribute to disease research [2]. Accordingly, how to identify these key genes from the massive high-dimensional genomic data is a hotspot and nodus in research. Furthermore, studies have testified that these key genes are efficaciously extracted by embedding learning [3]. Furthermore, cluster analysis is based on the similarity of each data point to classify the samples or genes, which is helpful for accurate determination of the cancer subtype. Some studies have also demonstrated that embedding learning and sparse regression is good for cluster analysis and feature selection [4,5].

Feature selection is used to pick out k features from m dimensional data (m > k) to optimize a specific index of the system [6]. Initially, some information on characteristic genes is extracted to constitute the data after the reduction to achieve dimensionality reduction. Following this, some genes associated with disease are dredged up from the low-dimensional data on medical research. At present, feature selection is extensively and continually studied owing to the usefulness and practicality of this method. However, traditional feature selection approaches have some issues: (1) The manifold structure is not fully considered, which can reflect the internal geometric structure in data [7]; and (2) they only use the statistical strategy, which can affect the accuracy and reliability of the results. Moreover, procedures of traditional feature selection method are performed independently, such as embedding learning and sparse regression [8,9]. However, a better performance can be achieved by combining two of the above independent procedures. Hou et al. came up with a new method of feature selection via Joint Embedding Learning and Sparse Regression (JELSR), according to the above ideas [5]. This method is a good solution to the above-mentioned issues. JELSR has a good effect on feature selection outstrip of these traditional methods. Nevertheless, there is still a problem thereinafter; since the $L_{2,1}$-norm penalty only sparsely constrains the rows of data, the sparsity of the method is far from satisfactory. However, if the sparseness is not enough, taking too many unrelated genes into account can cause serious errors. Hence, an efficient sparse method should be explored to strengthen the previous method.

LASSO (Least Absolute Shrinkage and Selection Operator) was first proposed by Robert Tibshirani [10]. It constructs an $L_{1}$-norm penalty function to obtain a more refined model and compresses some coefficients to get zero coefficients. Both the $L_{1}$-norm constraint and the $L_{2,1}$-norm constraint can produce sparse effects, but the sparsity of the $L_{1}$-norm constraint is decentralized and the $L_{2,1}$-norm constraint can only produce row sparseness, as shown in Figure 1. However, the combination of the $L_{1}$-norm and the $L_{2,1}$-norm constraints can generate internal row sparsity and enhance the correlation between rows and columns of the matrix [11], as shown in Figure 1, so the occurrence of the redundant and irrelevant genes can be reduced. In this paper, to obtain more sparse effects, we propose a new method by adding an $L_{1}$-norm constraint on the sparse regression matrix based on the JELSR (LJELSR). First, the corresponding graph is constructed to depict the inherent structure of the data. Then, the graph matrix is embedded into the following steps: feature ranking and data regression. Owing to the similarity between data points described by the above constructed graph, the clustering effect is improved to some degree. Finally, to get more sparse effects, we combine embedding learning and sparse regression with $L_{1}$-norm through linear weighting to complete the corresponding feature selection and cluster analysis. 

The major merits of our work are shown below:
More zero values are produced by adding an $L_{1}$-norm constraint on the sparse regression matrix, such that we get more sparse results. The internal geometric structure of data is preserved along with dimensionality reduction by embedding learning to reduce the occurrence of inaccurate results. Although an exact solution cannot be obtained by our method, we provide a convergent iterative algorithm to get the optimal results.


The other parts of this paper are arranged as listed below. Section 2 details some comparative experiments and the analysis of experimental results from different datasets. Section 3 describes some related materials and presents the methodology of LJELSR. A conclusion of this paper is given in Section 4.

## 2. Results

The LJELSR, JELSR [5], ReDac [12], and SMART [11] are used to select differentially expressed genes and test the performance of the proposed method for different genomic data. Among them, the JELSR, ReDac, and SMART are used as comparison methods.

### 2.1. Datasets

To validate the effectiveness of our method, the LJELSR, JELSR, ReDac, and SMART methods are run on three datasets, including the ALL_AML, the colon cancer, and the ESCA datasets. The ALL_AML dataset includes acute lymphoblastic leukemia (ALL) and acute myelogenous leukemia (AML) [13], and ALL has also been divided into T cell subtypes and B cell subtypes. The colon cancer dataset is obtained by [14], to facilitate clustering, and the samples are organized into two categories, namely, diseased and normal samples. Additionally, the esophageal carcinoma dataset (ESCA) is downloaded from the TCGA (The Cancer Genome Atlas, TCGA). The TCGA is a publicly available dataset, where it is acquired from https://tcgadata.nci.nih.gov/tcga/. Some details of the three datasets are listed in Table 1.

### 2.2. Parameters Selection

In our method, there are mainly four parameters involved, namely three balance parameters $\alpha_{1},\;\alpha_{2},\;\beta \;$ and the nearest neighbor number q. In ALL_AML, we select parameters $\alpha_{1}{,\;\alpha }_{2}$ and β from range $\left\{10^{5}{,\;10}^{7}{,\;10}^{9}{,\;10}^{11}{,\;10}^{13}{,\;10}^{15}{,\;10}^{17}{,\;10}^{19}{,\;10}^{21}{,\;10}^{23}{,\;10}^{25}\right\}$; in colon, $\alpha_{1}{,\;\alpha }_{2}$ and β are found from $\left\{10^{-5}{,\;10}^{-4}{,\;10}^{-3}{,\;10}^{-2}{,\;10}^{-1}{,\;10}^{0}{,\;10}^{1}{,\;10}^{2}{,\;10}^{3}{,\;10}^{4}{,\;10}^{5}\right\}$; and in ESCA, we choose parameters $\alpha_{1}{,\;\alpha }_{2}$ and β in the range of $\left\{10^{0}{,\;10}^{1}{,\;10}^{2}{,\;10}^{3}{,\;10}^{4}{,\;10}^{5}{,\;10}^{6}{,\;10}^{7}{,\;10}^{8}{,\;10}^{9}{,\;10}^{10}\right\}$. The above optimal parameters under different datasets are obtained by five-fold cross-validation. Besides, according to the existing literature [15,16] and a large number of experiments, when q is taken as 5 or 6, the experimental effect is better. In our experiment, we set the value of q to 5.

### 2.3. Evaluation Metrics

In this study, there are two metrics employed to assess all algorithms: *p*-value and clustering accuracy (ACC) [17]. Firstly, the size of the *p*-value is closely related to the relationship between the selected genes and the disease, and there is a negative correlation between them. Additionally, the *p*-value is obtained by the ToppFun tool, which is a public gene list functional enrichment analysis tool. The *p*-value cutoff is set as 0.01 in the whole experiment. Secondly, the level of ACC indicates the degree of excellence of the algorithm, and there is a positive correlation between them. The value of the ACC is obtained by the following formula:(1)$$\mathrm{ACC}=\frac{\sum_{i=1}^{n}\delta \left(s_{i},\mathrm{map}\left(c_{i}\right)\right)}{n}$$
where $c_{i}$ means the label of the clustering and $s_{i}$ is the label of the original data $x_{i}$. In addition, the value of the δ(x,Y) is 1 if x=y and is 0 otherwise, and map (·) is a mapping function. 

### 2.4. Feature Selection Analysis

#### 2.4.1. Experimental Results and Analysis on ALL_AML Dataset

For the sake of fairness, the LJELSR, JELSR, ReDac, and SMART are respectively used to extract 100 differentially expressed genes from the ALL_AML dataset to analyze their performance. To verify the effectiveness of the algorithm, the selected genes are put into ToppFun to get the *p*-values and the resulting *p*-values are arrayed in an ascending sort order. Then, we pick out the first ten terms listed in Table 2. The ten portions in bold font represent the best *p*-values. It can be seen from Table 2 that the *p*-values obtained by the LJELSR method are lower than the *p*-values obtained by the other three methods. Hence, the performance of the LJELSR surpasses the other three methods.

To further illustrate the relationship between the selected genes and ALL_AML, the selected differentially expressed genes are put in GeneCards for testing. GeneCards is a synthesis database of human genes, providing relationships between disease, gene expression, gene function, and so on. The top five differentially expressed genes associated with ALL_AML obtained by the LJELSR method are listed in Table 3 and the official names of these genes and related diseases are also listed. As can be seen from Table 3, the official name of the “*CD7*” is the “*CD7 Molecule*”. *CD7* is originally found in T cells of acute lymphoblastic leukemia [18]. It is a membrane glycoprotein of human T lymphocytes and thymocytes. Additionally, it plays an essential role in detecting the interactions of T-cells or B-cells during early lymphoid development. Thence, the loss of *CD7* can affect the expression of T-cells, which has a great impact on T-cell leukemia [19]. The “*MYB*” homologous official name is “*MYB*
*Proto-Oncogene, Transcription Factor*”. From the official name of this gene, *MYB* is not only a pro-oncogene, but also a factor that affects the transcription of genes. Furthermore, its duplication can cause leukemia [20]. Other genetic analyses are similar to the above analysis. In all, these genes are directly or indirectly related to leukemia. This table only simply displays partial functions of some of these genes.

#### 2.4.2. Experimental Results and Analysis on Colon Cancer Dataset

In this section, the analytical approach and procedure of this dataset are the same as the previous dataset. The 100 genes extracted by each method are tested for the Gene Ontology (GO) detection tool—ToppFun. The *p*-values gained by four methods are arranged in ascending order. We single out the *p*-values of the first ten items and list them in Table 4. The best *p*-values are indicated in bold typeface. From Table 4, it is obvious that the *p*-values obtained by the LJELSR method are smaller than the *p*-values obtained by the other three methods. Therefore, the performance of our method is a big plus over the other three methods.

In addition, the selected differentially expressed genes by the LJELSR method are put in GeneCards for testing. We choose the top five genes associated with colon cancer obtained by the LJELSR method to list in Table 5. Additionally, Table 5 also shows how these genes correspond to the official names and related diseases. From Table 5, the official name of the “*ACTB*” is “*Actin Beta*”. This gene can encode one of six different actin proteins, and change in the protein brought about by changes in the gene. It can affect certain biological processes. Furthermore, *ACTB* goes hand in hand with many cancers and plays a major role in lung and colorectal cancer, and so on [21]. Andersen et al. found that colon cancer is affected by *AC**TB* [22]. The “*WW Domain Containing Oxidoreductase*” is abbreviated as the “*WWOX*”. Different from the traditional tumor suppressor genes, its effect is more complicated and extensive on cellular function. Besides, the expression level of *WWOX* is different in two different colon cancer cell lines, which are the *HT29* and *SW480* cell lines [23]. Moreover, the expression of *WWOX* can lead to apoptosis, while defects in this gene are associated with multiple types of cancer. Analysis of the remaining genes is similar to the above-mentioned genes. Table 5 only shows some descriptions of partial genes associated with colon cancer.

#### 2.4.3. Experimental Results and Analysis on ESCA Dataset

In this subsection, the dataset we use is ESCA, which is different from the above two datasets. To further confirm the effectiveness of the algorithm, the experiment is run with this data and the selected differentially expressed genes are placed in ToppFun for GO analysis. We rank all the *p*-values in ascending order and choose first ten *p*-values to list in the Table 6, where the best results are highlighted. From Table 6, we can conclude that the *p*-values obtained by the LJELSR method are mostly smaller than the *p*-values obtained by the other three methods. Therefore, on the whole, the performance of the LJELSR outperforms the other three methods.

Additionally, the selected differentially expressed genes are put in GeneCards for testing. The top five genes associated with ESCA obtained by the LJELSR method, their name, and related diseases are all displayed in Table 7. From Table 7, the “*ERBB2*” corresponds to the official name “*Erb-B2 Receptor Tyrosine Kinase 2*”. *ERBB2* is the membrane receptor of 185kDa encoded by proto-oncogene *ERBB-2*, and is one of the members of the epidermal growth factor receptor family. Additionally, it has been verified that its amplification is closely related to the occurrence of esophageal cancer [24]. “*KRT5*” is the abbreviation of “*Keratin 5*”, and it is a member of the keratin family of momentous gene families that encodes the corresponding protein. Additionally, its changes affect the expression of this gene’s families, causing some complex diseases. For example, mutations in *KRT5* and *KRT14* can cause epidermolysis bullosa to a large extent [25]. The analysis of other genes is similar to the above analysis. Table 7 only shows some functions of partial genes, and detailed information on the remaining genes can be obtained from GeneCards.

#### 2.4.4. Differentially Expressed Genes Comparing by Methods

In this subsection, the selected differentially expressed genes are further analyzed. For the three datasets mentioned above, we explore the common differentially expressed genes and the unique differentially expressed genes obtained by different methods for the same dataset. We select 100 genes for each algorithm and pair them with the officially published disease-causing gene pool to obtain the verified genes. This disease-causing gene pool can be downloaded directly from GeneCards. Table 8 shows the common and unique differentially expressed genes obtained by the four methods for the ESCA dataset. The bold italic indicates the common differentially expressed genes excavated by the four methods. The underlined italic indicates the unique differentially expressed genes excavated by LJELSR, which are not found by other methods. The “Number” means the total number of verified genes. As can be seen from the table, LJELSR has selected more proven genes than other methods; *ANXA1*, *MUC6*, *FN1*, *PKM*, *CD24*, *GLUL*, *PLEC*, *PIGR*, *ACTB*, *PABPC1*, *LYZ*, and *SPRR1B* are the common differentially expressed genes; the unique differentially expressed genes of the LJELSR method are *FSCN1*, *ITGB4*, *LAMC2*, *HLA-B*, *LAMB3*, *HLA-C*, *SLC7A5*, *ENO1*, and so on. These genes play an important role in studying the relationship between genes and diseases, the details of these genes can be obtained from GeneCards, and some of the genes have been explained above. In summary, LJELSR has an advantage over other methods in that it can mine more and more valuable differentially expressed genes.

### 2.5. Clustering Analysis

Kmeans is one of simple algorithms that solve the clustering problem, and its procedure is a relatively easy and simple way to classify the samples by setting a certain number of clusters in advance [26]. For all methods, different datasets have different numbers of sample clustering. The Kmeans method is used to cluster samples on the ALL_AML, colon cancer, and ESCA datasets, where their number of sample clustering is three, two, and two, respectively. Different methods are run for different data to get the values of ACC, and the details are shown in Table 8, where the largest values are marked in bold typeface. From Table 9, we can conclude that the values of ACC obtained by the LJELSR method for different datasets are almost all higher than the other methods. In this section, based on the above analysis, we can summarize that the efficiency of our method is higher than other methods.

## 3. Materials and Methods

### 3.1. Related Notations and Definitions

In this paper, $m^{i}$ and $m_{j}$ are defined as the *i*-th row and *j*-th column of matrix M, respectively. Denote $X\in R^{m\times n}$ as the original data matrix, where its row represents a gene (feature) and its column represents a sample. For an arbitrary matrix M, $L_{r,s}$ norm is represented as below: (2)$${\left\|M\right\|}_{r,s}={\left(\overset{n}{\underset{i=1}{\sum }}{\left(\overset{m}{\underset{j=1}{\sum }}{\left|m_{ij}\right|}^{r}\right)}^{\frac{s}{r}}\right)}^{\frac{1}{s}}$$
where r and s represent a positive number. When r=s=1, the above formula becomes the expression of the $L_{1}$ norm or LASSO, i.e.,
(3)$${\left\|M\right\|}_{1}=\overset{n}{\underset{i=1}{\sum }}\overset{m}{\underset{j=1}{\sum }}\left|m_{ij}\right|$$

Similarly, when r=2, s=1, and (2) becomes the expression of the $L_{2,1}$ norm, i.e.,
(4)$${\left\|M\right\|}_{2,1}=\overset{n}{\underset{i=1}{\sum }}{\left(\overset{m}{\underset{j=1}{\sum }}{\left|m_{ij}\right|}^{2}\right)}^{\frac{1}{2}}$$

Additionally, the inherent geometric structure of data is fully taken into account to deal with data for lessening the appearance of the inexact results in actual applications. Considering its benefits, it is also added to our work. Therefore, we firstly construct a q -nearest-neighbor graph G with n vertices in the data space, where a vertex corresponds to a data point [15], and q is the nearest neighbor number. $w_{ij}$ represents the correlation between two data point $x_{j}$ and $x_{j}$. All $w_{ij}$ make up the weight matrix W. There are many strategies to compute W. However, three strategies are frequently employed [15].

(1) 0-1 weighting: (5)$$w_{ij}=\{\begin{array}{cc} 1, & {if\;x}_{i}\in N_{q}\left(x_{j}\right){\;or\;x}_{j}\in N_{q}\left(x_{i}\right)\\ 0,\; & otherwise \end{array}$$
where $N_{q}\left(x_{i}\right)$ indicates the set gained by the way of q -nearest neighbors of the data point $x_{i}$.

(2) Heat kernel weighting:(6)$$w_{ij}=\{\begin{array}{cc} e^{-\frac{\|x_{i}{-x}_{j}\|{}^{2}}{\sigma }}, & {if\;x}_{i}\in N_{q}\left(x_{j}\right){\;or\;x}_{j}\in N_{q}\left(x_{i}\right)\\ 0,\; & otherwise \end{array}$$
where σ is a proper constant, and its value is obtained by the previous experience.

(3) Dot-product weighting: (7)$$w_{ij}=\{\begin{array}{cc} x_{i}{}^{T}x_{j}, & {if\;x}_{i}\in N_{q}\left(x_{j}\right){\;or\;x}_{j}\in N_{q}\left(x_{i}\right)\\ 0,\; & otherwise \end{array}$$

These three strategies are applied to different occasions. Since the operation of the 0-1 weighting is relatively uncomplicated, it is commonly used for computing the weight matrix. The second is widely applied to image data, and the third is frequently used in the IR community for processing documents [15].

Next, we define a diagonal matrix D, where its diagonal values are given as $d_{ii}=\underset{j}{\sum }w_{ij}$. The graph Laplacian matrix L is defined as L=D−W [27].

### 3.2. Joint Embedding Learning and Sparse Regression (JELSR)

Traditional feature selection methods are performed independently. To further heighten the performance of the previous algorithm, the JELSR method was proposed by Hou et al. [5]. Firstly, it uses linear approximation weights and the $L_{2,1}$-norm regularization to combine embedding learning and sparse regression to establish a new objective function. Then, the sparse regression matrix is used to finish the corresponding feature selection [5]. The objective function of the JELSR algorithm is written as follows:(8)$$\begin{array}{c} \underset{P,Y}{\min}\mathrm{tr}\left({\mathrm{YLY}}^{T}\right)+\beta \left({\left\|P^{T}X-Y\right\|}_{2}^{2}{+\alpha \|P\|}_{2,1}\right)\\ s.t{.\mathrm{YY}}^{T}{=I}_{k\times k} \end{array}$$
where $Y\in R^{k\times n}$ is a low dimensional embedding matrix; $P\in R^{m\times k}$ is the sparse regression matrix; α and β are two balance parameters; and k is the dimension of low-dimensional space.

### 3.3. The Proposed Method

Assuming that the expression level of the gene is within the normal range, the higher the sparseness of the sparse regression matrix, the easier it is to find the differentially expressed genes. What is more, the results of identifying the differentially expressed genes are much more accurate.

Compared with the previous methods, the JELSR method proposed by Hou et al. [5] has a good effect on the feature selection. However, it is inevitable that some redundancy values and artificial noise values in the sparse regression matrix have to be taken into account, such that the sparseness of this method is far from satisfactory. Consequently, it is necessary to discover an efficacious sparse method to improve the performance. Therefore, we propose the LJELSR method, which may improve the sparseness of the algorithm and the accuracy of the results.

The objective function of LJELSR is:(9)$$\begin{array}{c} \underset{P,Y}{\min}\mathrm{tr}\left({\mathrm{YLY}}^{T}\right)+\beta ({\left\|P^{T}X-Y\right\|}_{2}^{2}{+\alpha }_{1}{\left\|P\right\|}_{1}{+\alpha }_{2}{\left\|P\right\|}_{2,1})\\ s.t{.\mathrm{YY}}^{T}{=I}_{k\times k} \end{array}$$
where $Y\in R^{k\times n}$ is a low-dimensional embedding matrix; $P\in R^{m\times k}$ is the sparse regression matrix; and $\alpha_{1}$, $\alpha_{2}$, and β are three balance parameters.

### 3.4. Optimization

Since there are the $L_{1}$-norm and the $L_{2,1}$-norm constraints on the objective formula, it is difficult to optimize and solve the optimal solution directly. With the elicitation of Wang et al. [11] and Nie et al. [28], the iterative strategy is introduced to solve the above problem. Now, we will explain in detail the specific optimization process of our method. 

Before solving the optimization problem, we introduce two diagonal matrices, $U\in R^{m\times m}$ and $\tilde{U}\in R^{m\times m}$, whose *i*-th diagonal values are defined as follows:(10)$$U=\overset{k}{\underset{i=1}{\sum }}U_{i},\;{\left(U_{i}\right)}_{jj}=\frac{1}{2\left|P_{ji}\right|}\left(j=1\ldots m\right)$$
(11)$${\tilde{U}}_{jj}=\frac{1}{2{\left\|p^{j}\right\|}_{2}}$$
where $U_{i}$ represents *i*-th diagonal matrix U; ${\left(U_{i}\right)}_{jj}$ represents the *j*-th diagonal value of the *i*-th matrix U; and ${\tilde{U}}_{jj}$ indicates the *j*-th diagonal value of matrix $\tilde{U}$.

To prevent the emergence of spillover, a small constant ε is added to the diagonal matrices  U and $\tilde{U}$, respectively, that is:(12)$$U=\overset{k}{\underset{i=1}{\sum }}U_{i},\;{\left(U_{i}\right)}_{jj}=\frac{1}{2\max\left(\left|p_{ji}\right|,\varepsilon \right)}\left(j=1\ldots m\right)$$
(13)$${\tilde{U}}_{jj}=\frac{1}{2\max\left({\left\|p^{j}\right\|}_{2},\varepsilon \right)}$$

Owing to the fact that the partial derivatives of ${\left\|P\right\|}_{1}$ and $\mathrm{tr}\left(P^{T}\mathrm{UP}\right)$ on P are identical, ${\left\|P\right\|}_{1}$ can be replaced by $\mathrm{tr}\left(P^{T}\mathrm{UP}\right)$. Analogously, we carry out an exchange for ${\left\|P\right\|}_{2,1}$ and $\mathrm{tr}\left(P^{T}\tilde{U}P\right)$. Therefore, (9) can be rewritten as:(14)$$\begin{array}{c} \underset{P,U,\tilde{U},Y}{\min}\mathrm{tr}\left({\mathrm{YLY}}^{T}\right)+\beta \left({\left\|P^{T}X-Y\right\|}_{2}^{2}+\alpha_{1}\mathrm{tr}\left(P^{T}\mathrm{UP}\right)+\alpha_{2}\mathrm{tr}\left(P^{T}\tilde{U}P\right)\right)\\ s.t.{\mathrm{YY}}^{T}{=I}_{k\times k} \end{array}$$

We firstly optimize the matrix P. We denote
(15)$$L\left(P,U,\tilde{U}\right)={\left\|P^{T}X-Y\right\|}_{2}^{2}+\alpha_{1}\mathrm{tr}\left(P^{T}\mathrm{UP}\right)+\alpha_{2}\mathrm{tr}\left(P^{T}\tilde{U}P\right)$$
When two diagonal matrices U and $\tilde{U}$ are fixed, we compute the partial derivative of $L\left(P,U,\tilde{U}\right)$ on P and make it equal to zero. Therefore, we can get the following equation: (16)$$\frac{\partial L\left(P\right)}{\partial P}{=2\mathrm{XX}}^{T}P-{2\mathrm{XY}}^{T}+{2\alpha }_{1}\mathrm{UP}+{2\alpha }_{2}\tilde{U}P=0$$
namely,
(17)$$P={\left({\mathrm{XX}}^{T}{+\alpha }_{1}{U+\alpha }_{2}\tilde{U}\right)}^{-1}{\mathrm{XY}}^{T}$$

To facilitate optimization, we introduce an auxiliary variable A into the objective formula, and denote $A={\mathrm{XX}}^{T}{+\alpha }_{1}{U+\alpha }_{2}\tilde{U}$. According to the above analysis, (17) is brought into (14), such that we will get
(18)$$L\left(P,U,\tilde{U},Y\right)=\mathrm{tr}\left(Y\left(L+{\beta I}_{n\times n}-{\beta X}^{T}A^{-1}X\right)Y^{T}\right)$$

Since Y is subject to the orthogonal constraint ${\mathrm{YY}}^{T}{=I}_{k\times k}$, the optimization problem of Y becomes
(19)$$\begin{array}{c} \arg\;\underset{Y}{\min}\mathrm{tr}\left(Y\left({L+\beta I}_{n\times n}{-\beta X}^{T}A^{-1}X\right)Y^{T}\right)\\ s.t{.\mathrm{YY}}^{T}{=I}_{k\times k} \end{array}\;$$
When A and L are fixed, we use the strategy of the eigen-decomposition of the matrix G = $\left({L+\beta I}_{n\times n}{-\beta X}^{T}A^{-1}X\right)Y^{T}$ to update Y in (19). For $Y_{i}\left(i=1,2,\cdots ,k\right)$, we firstly choose the k smallest eigenvalues of the matrix G, and then seek out the corresponding eigenvectors to constitute a new matrix $Y=\left(Y_{1}{,Y}_{2},\cdots {,Y}_{k}\right)$ [29]. In addition, the diagonal matrices U and $\tilde{U}$ are updated by (12) and (13) when P is fixed, respectively. Additionally, we initialize U and $\tilde{U}$ as an identity matrix, respectively.

### 3.5. Feature Selection

According to the above update rules, the sparse regression matrix P is acquired after repeated iterations of LJELSR. Then, to acquire some differentially expressed genes, we conduct a detailed analysis of the matrix P. In the first place, each element of the matrix P is subjected to absolute processing. Secondly, we sum the absolute values by each row of the sparse regression matrix P and get a new vector, as follows.
(20)$$\bar{P}={\left({\bar{P}}_{1},{\bar{P}}_{2},\cdots ,{\bar{P}}_{m}\right)}^{T}$$
(21)$${\bar{P}}_{i}=\sum_{j=1}^{k}=|{\bar{P}}_{ij}|$$

Then, we rank the vector ${\bar{P}}_{i}$ in descending order to get a new vector, as follows:(22)$$\tilde{P}={\left({\tilde{P}}_{1},{\tilde{P}}_{2},\cdots ,{\tilde{P}}_{m}\right)}^{T}$$

Finally, the genes corresponding to the first l values are selected as differentially expressed genes to analyze their property (l < m). By and large, the value of the element is directly proportional to the importance of the corresponding gene. Next, we put the selected genes into ToppFun and GeneCards to analyze them, where they are publicly accessible from https://toppgene.cchmc.org/enrichment.jsp and http://www.genecards.org/, respectively.

In summary, the procedure of LJELSR is shown in Algorithm 1. 

**Algorithm 1.** Procedure of LJELSR.**Input:** Data matrix X; Neighborhood size q; Balance parameters α1,α2,β; Dimensionality of embedding k; Feature selection number l. **Output:** Selected feature index set {P1,P2,⋯,Pm}
Stage one: Graph construction Construct the weight matrix W; Compute the diagonal matrix D, graph Laplacian matrix L; Stage two: Alternative optimization, Initialize U=U˜=Im×m; Loop Update Y and fix A, L by (19), Update P and fix U, U˜ by (17), Update U, U˜ and fix  P by (12) and (13). until convergenceStage three: Feature selection


### 3.6. Convergence Analysis

In this study, an alternative algorithm is utilized to finish iteratively updating work of the proposed method. Now, let us analyze its convergence behavior of LJELSR. The lemma given below was proposed by Nie et al. [28].

**Lemma** **1.**
*For any non-zero vectors ${P,P}_{t}\in R^{k}$, the following inequality holds:*
(23)$${\left\|P\right\|}_{2}-\frac{{\left\|P\right\|}_{2}^{2}}{2\|P_{t}\|{}_{2}}\leq \|P_{t}\|{}_{2}-\frac{\|P_{t}\|{}_{2}^{2}}{{2\|P}_{t}\|{}_{2}}$$


The convergence result of the proposed method is explained by the following theorem:
**Theorem** **1.***The value of the target function for each iteration is monotonically decreasing in the Algorithm 1. Detailed proof of Theorem 1 is given in the Appendix A*.

## 4. Conclusions

In this paper, we discuss a new feature selection method by adding an $L_{1}$-norm constraint on the sparse regression matrix based on the JELSR method. Firstly, the four methods are executed to select differentially expressed genes and cluster the samples from ALL_AML, colon cancer, and ESCA datasets, respectively. Secondly, some of materials related to this paper are presented, and our methods and the corresponding optimization strategy are given. Finally, the conclusion is drawn that the performance of the proposed method is better than other methods through the experimental results. 

## Figures and Tables

**Figure 1 ijms-20-00886-f001:**
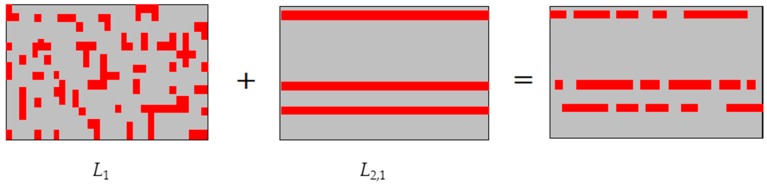
The diagrammatic sketch of different norms.

**Table 1 ijms-20-00886-t001:** Details of the three datasets.

Datasets	Genes	Samples	Classes	Description
ALL_AML	5000	38	3	acute lymphoblastic leukemia and acute myelogenous leukemia
colon	2000	62	2	colon cancer
ESCA	20,502	192	2	esophageal carcinoma

**Table 2 ijms-20-00886-t002:** The *p*-values of different methods applied to the ALL_AML dataset.

ID	LJELSR	JELSR	ReDac	SMART
GO:0006955	2.249 × 10^−20^	3.157 × 10^−18^	8.447 × 10^−14^	1.900 × 10^−10^
GO:0050776	8.768 × 10^−18^	1.312 × 10^−15^	7.228 × 10^−1^^2^	1.000 × 10^−9^
GO:0045321	1.306 × 10^−16^	1.808 × 10^−14^	1.154 × 10^−11^	2.500 × 10^−10^
GO:0001775	1.548 × 10^−15^	1.604 × 10^−13^	6.621 × 10^−11^	1.590 × 10^−10^
GO:0051251	2.005 × 10^−15^	4.157 × 10^−14^	9.117 × 10^−12^	4.380 × 10^−12^
GO:0007159	2.098 × 10^−15^	2.754 × 10^−15^	3.712 × 10^−11^	1.510 × 10^−10^
GO:0002682	2.477 × 10^−15^	2.725 × 10^−14^	7.817 × 10^−12^	5.870 × 10^−8^
GO:0046649	3.964 × 10^−15^	5.426 × 10^−14^	3.164 × 10^−10^	1.540 × 10^−11^
GO:0016337	7.034 × 10^−15^	8.993 × 10^−14^	5.253 × 10^−11^	1.940 × 10^−11^
GO:0070486	7.220 × 10^−15^	9.350 × 10^−15^	1.299 × 10^−11^	5.400 × 10^−10^

**Table 3 ijms-20-00886-t003:** The top five genes selected by LJELSR for the ALL_AML dataset.

Gene	Gene Official Name	Related Diseases
*CD34*	*CD34 Molecule*	Dermatofibrosarcoma Protuberans and Hypercalcemic Type Ovarian Small Cell Carcinoma
*CD7*	*CD7 Molecule*	Pityriasis Lichenoides Et Varioliformis Acuta and T-Cell Leukemia
*MYB*	*MYB Proto-Oncogene, Transcription Factor*	Acute Basophilic Leukemia and Angiocentric Glioma
*CXCR4*	*C-X-C Motif Chemokine Receptor 4*	Whim Syndrome and Human Immunodeficiency Virus Infectious Disease
*CTSG*	*Cathepsin G*	Papillon-Lefevre Syndrome and Cutaneous Mastocytosis

**Table 4 ijms-20-00886-t004:** The *p*-values of different methods for the colon dataset.

ID	LJELSR	JELSR	ReDac	SMART
GO:0006614	1.612 × 10^−17^	9.836 × 10^−14^	2.677 × 10^−14^	1.016 × 10^−12^
GO:0006613	3.970 × 10^−17^	2.060 × 10^−13^	5.617 × 10^−14^	1.970 × 10^−12^
GO:0045047	5.074 × 10^−17^	2.519 × 10^−13^	6.872 × 10^−14^	2.360 × 10^−12^
GO:0072599	8.161 × 10^−17^	3.720 × 10^−13^	1.016 × 10^−13^	3.348 × 10^−12^
GO:0022626	3.104 × 10^−16^	8.999 × 10^−13^	2.465 × 10^−13^	1.125 × 10^−11^
GO:0000184	3.751 × 10^−16^	1.301 × 10^−12^	3.568 × 10^−13^	1.029 × 10^−11^
GO:0003735	5.428 × 10^−16^	6.787 × 10^−13^	1.412 × 10^−13^	3.310 × 10^−12^
GO:0070972	6.180 × 10^−16^	1.960 × 10^−12^	5.384 × 10^−13^	1.488 × 10^−11^
GO:0019083	9.571 × 10^−16^	1.932 × 10^−12^	1.547 × 10^−11^	3.005 × 10^−10^
GO:0044445	1.372 × 10^−15^	1.487 × 10^−12^	3.106 × 10^−13^	8.772 × 10^−12^

**Table 5 ijms-20-00886-t005:** The top five genes selected by LJELSR for the colon dataset.

Gene	Gene Official Name	Related Diseases
*MUC3A*	*Mucin 3A, Cell Surface Associated*	Cap Polyposis and Hypertrichotic Osteochondrodysplasia
*ACTB*	*Actin Beta*	Dystonia, Juvenile-Onset and Baraitser-Winter Syndrome 1
*WWOX*	*WW Domain Containing Oxidoreductase*	Spinocerebellar Ataxia, Autosomal Recessive 12andEpileptic Encephalopathy, Early Infantile, 28
*SPI1*	*Spi-1 Proto-Oncogene*	Inflammatory Diarrhea and Interdigitating Dendritic Cell Sarcoma
*RPS24*	*Ribosomal Protein S24*	Diamond-Blackfan Anemia 3 and Diamond-Blackfan Anemia

**Table 6 ijms-20-00886-t006:** The *p*-values of different methods for the ESCA dataset.

ID	LJELSR	JELSR	ReDac	SMART
GO:0005198	2.772 × 10^−^^27^	8.941 × 10^−^^18^	1.096 × 10^−^^18^	1.036 × 10^−^^4^
GO:0070161	7.504 × 10^−^^21^	3.347 × 10^−^^14^	2.733 × 10^−^^23^	6.527 × 10^−^^14^
GO:0030055	1.400 × 10^−^^19^	1.111 × 10^−^^10^	3.576 × 10^−^^17^	1.619 × 10^−^^12^
GO:0005912	6.655 × 10^−^^19^	1.946 × 10^−^^11^	4.401 × 10^−^^20^	3.498 × 10^−^^12^
GO:0005925	1.319 × 10^−^^18^	7.671 × 10^−^^11^	2.103 × 10^−^^17^	1.061 × 10^−^^12^
GO:0005924	1.746 × 10^−^^18^	9.246 × 10^−^^11^	2.747 × 10^−^^17^	1.313 × 10^−^^12^
GO:0005615	5.538 × 10^−^^16^	8.395 × 10^−^^20^	3.985 × 10^−^^15^	2.117 × 10^−^^17^
GO:0030054	5.243 × 10^−^^14^	4.156 × 10^−^^10^	5.243 × 10^−^^14^	4.510 × 10^−^^9^
GO:0005200	3.076 × 10^−^^13^	1.301 × 10^−^^10^	2.062 × 10^−^^13^	2.358 × 10^−^^4^
GO:0042060	7.442 × 10^−^^13^	2.650 × 10^−^^9^	4.536 × 10^−^^12^	8.790 × 10^−^^11^

**Table 7 ijms-20-00886-t007:** The top five genes selected by LJELSR for the ESCA dataset.

Gene	Gene Official Name	Related Diseases
*ERBB2*	*Erb-B2 Receptor Tyrosine Kinase 2*	Glioma Susceptibility 1andOvarian Cancer, Somatic
*KRT14*	*Keratin 14*	Epidermolysis Bullosa Simplex, Koebner Type and Epidermolysis Bullosa Simplex, Recessive 1
*KRT5*	*Keratin 5*	Epidermolysis Bullosa Simplex, Dowling-Meara Type and Epidermolysis Bullosa Simplex, Weber-Cockayne Type
*KRT19*	*Keratin 19*	Anal Canal Adenocarcinoma and Thyroid Cancer
*KRT4*	*Keratin 4*	White Sponge Nevus 1andWhite Sponge Nevus Of Cannon, Krt4-Related

**Table 8 ijms-20-00886-t008:** Differentially expressed genes of four methods for the ESCA dataset.

Methods	Differentially Expressed Genes	Number
LJELSR	***ERBB2, KRT14, KRT5, KRT19, KRT4, TFF1,*** *FSCN1* ***, KRT13,*** *ITGB4* ***, ANXA1, MUC6,*** *LAMC2* ***,*** *HLA-B* ***, KRT16, JUP, KRT17,*** *LAMB3* ***, ATP4A, DSP,*** *LAMA3* ***, FOS, FN1, CTSB, MYH11,*** *HLA-C* ***, LDHA, PKM,*** *SLC7A5* ***, PSCA, SERPINA1, S100A7, S100A9, CRNN, S100A8, B2M, DMBT1, CD24,*** *ENO1* ***, TNC,*** *KRT15* ***, GLUL,*** *HSPA1A* ***, NDRG1, LCN2,*** *COL17A1* ***,*** *CEACAM6* ***, REG1A, PLEC, GAPDH, PIGR, AGR2,*** *ANPEP* ***, PKP1, ACTB, FLNA,*** *PI3* ***, FTL, CTSE,*** ***PABPC1, PGC, ALDOA, EEF2,*** *KRT6B* ***, LYZ, CLDN18, SPRR1B,*** *KRT6C* ***,*** *PPP1R1B* ***, PGA3,*** *COL3A1* ***,*** *C3* ***,*** *REG1B* ***,*** *PERP* ***, KRT6A,*** *PGA5* ***,*** *CES1* ***,*** *PGA4* ***,*** *EIF1*	78
JELSR	***MUC1, KRT14, KRT5, KRT19, KRT8, KRT4, TFF1, FN1, MUC6, S100A8, CTSD, SPRR3, ATP4A, HSPB1, FOS, CEACAM5, GJB2, H19, EZR, KRT16, KRT13, CTSB, JUP, ANXA1, TFF2, S100A9, SFN, KRT17, PSCA, S100A2, MUC5B, COL1A1, CD24, PKM, DSC3, GLUL, MALAT1, REG1A, ACTN4, DSG3, LCN2, DSP, S100A7, B2M, MYH9, PKP1, S100A11, PIGR, HSPA8, PLEC, TRIM29, EEF1A1, ATP1B1, AGR2, LYZ, ACTB, PGC, PABPC1, SPRR2A, SPRR1B, SPRR1A, CA2, REG4, P4HB, CLDN18, CTSE, EEF2, CREB3L1, KRT6A, A2M, PGA3***	71
ReDac	***KRT14, KRT5, KRT4, FN1, CRNN, TGM3, MUC6, S100A8, IL1RN, SPRR3, ATP4A, HSPB1, MAL, KRT16, KRT13, JUP, THBS1, ANXA1, S100A9, PSCA, ECM1, CD24, PKM, FTL, HSPG2, DES, GLUL, MALAT1, PPL, EMP1, ACTN4, MYH11, CSTA, GAPDH, TAGLN, DSP, B2M, MYH9, GSN, PKP1, S100A11, PIGR, HSPA8, FLNA, PLEC, MYLK, CSTB, TRIM29, EEF1A1, RPL3, LYZ, PSAP, ACTB, ALDOA, PGC, PABPC1, SYNM, SPRR2A, SPRR1B, SPRR1A, REG4, P4HB, EEF2, KRT6A, ACTG2, PGA3***	66
SMART	***ERBB2, CCND1, GSTP1, CD44, KRT19, MUC4, MUC2, GRB7, TFF1, HLA-A, FN1, SOD2, ITGA6, NDRG1, SPP1, SERPINA1, MUC6, CTSD, HSPB1, CEACAM5, H19, CTSB, F5, ITGB1, ANXA1, SDC1, DMBT1, CLU, LDHA, CD24, APP, PKM, FTL, HSPG2, TNC, GLUL, MALAT1, NTS, LCN2, MYH9, PIGR, FLNA, CD55, PLEC, TSPAN8, EEF1A1, AGR2, LYZ, GPX2, ACTB, DSG2, PABPC1, SPRR1B, REG4, SCD, CLDN18, FAT1***	57

**Table 9 ijms-20-00886-t009:** ACC of different methods for different datasets.

Methods	ALL_AML	Colon	ESCA
LJELSR	81.579	64.520	96.354
JELSR	81.579	61.290	95.833
ReDac	68.421	61.290	95.313
SMART	44.740	63.980	94.790
Kmeans	78.530	53.420	96.350

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
