# Peer review of "LJELSR: A Strengthened Version of JELSR for Feature Selection and Clustering"

_ijms, 2019, doi:10.3390/ijms20040886_

Reviewer 1 Report

The authors consider a version of Joint Embedding Learning and Sparse Regression within Tibshirani’s LASSO methodology. It is an interesting topic that deserves attention. I have several comments on the manuscript.

Line 9. Probably “segregate” is not the most appropriate word here. Rephrase.

Lines 10-11. Rephrase “JELSR is proposed via Joint Embedding Learning and Sparse Regression”. For example, it could be “Joint Embedding Learning and Sparse Regression (JELSR) was proposed.”

Lines 23-24. Is “data” plural or singular? The authors should be consistent. In Line 23 you have written “… data are …” while in Line 24 – “… data is …”.

Line 25. What do the authors mean by “the dimensionality disaster”? In addition, “slew” is not appropriate here.

Line 78. The purpose of Figure 1 is not clear. The authors should provide more explanations in the text (see Line 59 where you have mentioned the figure). Remove double full stop in Line 78.

Line 96. What are p and q? Are they positive?

Line 106. Probably “proverbially” is not the most appropriate word here. Rephrase.

Lines 138-141, Formulas (4) and (5). What are U_i and u_i? Provide more details. Explain in detail the matrices U and \tilde U.

Line 142. What do the authors mean by “the emergence of spillover”?

Lines 187-188. Algorithm 1. What is the stopping criterion in the algorithm?

Section 4.3. The authors compare the p-values. What about the power (as well as specificity and sensitivity) of the methods?

Author Response

Our response is in the uploaded file.

Reviewer 2 Report

Please find them in the attachment

Author Response

Our response is in the uploaded file.

Round  2

Reviewer 1 Report

The authors have made the necessary changes asked by the reviewers.The manuscript can be accepted for publication in IJMS. 

Reviewer 2 Report

I do not have any further comments.